# Evaluating the efficacy of intragastric botulinum toxin a injections with two different quantities and sites for obesity

Tien-Yow Chuang[1,2,3☯], Cheng-Hung Chiu[4☯], I-Wen Penn[5,6*], Cheng-Di Chiu[7,8,9,10*]

1 Department of Physical Medicine and Rehabilitation, China Medical University Hospital, Taichung, Taiwan, 2 Department of Physical Medicine and Rehabilitation, School of Medicine, China Medical University, Taichung, Taiwan, 3 Department of Physical Medicine and Rehabilitation, Taichung Municipal Geriatric Rehabilitation General Hospital, Taichung, Taiwan, 4 Department of Plastic and Aesthetic, Genesis Clinic, Taipei, Taiwan, 5 Division of Physical Medicine and Rehabilitation, Fu Jen Catholic University Hospital, New Taipei City, Taiwan, 6 School of Medicine, Fu Jen Catholic University, Taipei, Taiwan, 7 Department of Neurosurgery, China Medical University Hospital, Taichung, Taiwan, 8 Spine Center, China Medical University Hospital, Taichung, Taiwan, 9 School of Medicine, China Medical University, Taichung, Taiwan, 10 Graduate Institute of Biomedical Science, China Medical University, Taichung, Taiwan

☯ These authors have contributed equally to this work and share first authorship
* cdchiu4046@gmail.com

## Abstract

### Introduction

Intragastric injections of botulinum toxin A (BTX-A) have shown promise in aiding weight reduction among obese patients, with a favorable safety profile and minimal adverse effects; however, the inconsistent results from prior studies highlight the need to examine key factors in the research of intragastric injection of botulinum toxin A, such as the dosage of BTX-A, the number and placement of injections. This study examines the efficacy of varying high doses, multi-gastric sites botulinum toxin A injections for weight control.

### Materials and Methods

A total of 103 patients aged 18–65 with a BMI ≥ 25 kg/m² were assigned to four groups receiving endoscopic BTX-A injections at varying doses and sites: Group 1 (400 IU, fundus and body), Group 2 (300 IU, fundus and body), Group 3 (400 IU, antrum and body), and Group 4 (300 IU, antrum and body). Baseline comparisons used ANOVA, while a mixed model assessed the interaction among injection site, dose, and time on outcomes.

### Results

Baseline measures showed no group differences in bodyweight, BMI, or body fat. The mixed model indicated significant reductions in bodyweight, BMI, and body

**Data availability statement:** All relevant data are within the paper and its Supporting Information files.

**Funding:** This work was supported by China Medical University [CMU112-IP-02]. The funders had no role in study design, data collection and analysis, decision to publish, or preparation of the manuscript.

**Competing interests:** The authors have declared that no competing interests exist.

fat with gastric fundus and body injections. Site and dose interactions significantly affected bodyweight ($p = 0.024$) and body fat ($p = 0.041$), but not BMI.

## Conclusions

Endoscopic intragastric BTX-A injections effectively reduce body weight, BMI, and body fat, particularly with injections in the fundus and body regions.

---

## Introduction

Obesity develops gradually when the number of calories consumed surpasses the energy expended. This condition, known as energy imbalance, can be influenced by a variety of risk factors. Some of these factors are personal, such as knowledge, skills, and habits, while others are shaped by environmental conditions. [1] Recognizing the risk factors is essential for taking proactive steps toward achieving a healthy weight and minimizing the risk of obesity-related health issues, including heart disease, diabetes, hypertension, sleep apnea, and gastroesophageal reflux disease, all of which can negatively impact quality of life. [2,3]

Surgical, pharmacotherapeutic and/or behavioral interventions to regain a healthy weight and address comorbidities of obesity are not uncommon. [3–5] No generally accepted practical strategies have yet been established. The expectation that current treatment modalities cannot achieve healthy weight stems from the belief that lifestyle changes have limited long-term effects on weight loss maintenance, pharmacotherapy poses a risk of side effects, and bariatric surgery may lead to serious adverse events. [6,7]

What is the optimal method to manage overweight and obesity? Ongoing medical efforts to enhance or discover new treatments for mild obesity have resulted in several routes that clinicians can implement. In recent decades, promising developments have emerged in the endoscopic intragastric injection of botulinum toxin (IGIB) to control bodyweight, but they remain a challenge. [6–11] The first trial in humans showed an 8.9% reduction in body weight. [12] However, subsequent randomized controlled trials and systematic reviews yielded heterogeneous and conflicting results. [6,7] This board range of results may be because (a) some studies contained a small sample size and measured in different ways, (b) the periods of follow-up were different, (c) the variations in dosage, number of injections, and injection sites were the most significant confounding factors that must be carefully considered.

The underlying principle of IGIB is that inhibiting gut pacemaker activity by blocking acetylcholine-mediated motility in the gastric antrum and body can delay gastric emptying and enhance satiety. [7,8,13] Additionally, IGIB may lower the secretion of ghrelin, a hormone that stimulates appetite, in the gastric fundus, while also reducing gastric capacity, thereby promoting early satiety. [8,10,14]

However, there is no consensus within the field on how IGIB should be administered or with what doses and sites of injections. This study aims to evaluate the efficacy of injecting varying doses of botulinum toxin A (BTX-A) into different gastric sites for bodyweight control.

## Materials and methods

### Subjects

The study collected data from patients who received BTX-A injections at the Weight Control Center of the Genesis Clinic between October 2020 and December 2023. This retrospective observational study design was approved by the local institutional review board (Antai Medical Care Cooperation, Anti-Tian-Sheng Memorial Hospital) for human research and adhered to the Occupational Health and Safety Administration regulations (24–116-C). Informed consent was waived in our study. The data used in this study were accessed for research purposes on 04/12/2024. During and after data collection, one of the coauthors had access to information that could identify individual participants.

Inclusion criteria included age between 18 and 65 years with a body mass index (BMI) of at least 25 kg/m². A BMI between 25.0 and 29.9 is classified as overweight, while a BMI of 30.0 or above is considered obesity. There were 17 patients (17/33) in Group 1, 21 patients (21/46) in Group 2, 3 patients (3/5) in Group 3, and 4 patients (4/19) in Group 4 were classified as obesity, while the other patients in each group were considered overweight. Moreover, obesity-related complications encompassed type 2 diabetes mellitus, hypertension, and obstructive sleep apnea. Patients who had history of cancer, current or potential pregnancy, previous gastric surgery, neuromuscular disorders, gastrointestinal diseases, or loss of follow-up were excluded. All participants were consecutively recruited and assigned to each group. They underwent blood biochemistry tests and bioelectrical impedance analysis (BIA) for body composition before and after IGIB. Additionally, a total esophagogastroduodenoscopy was done before the application of IGIB.

### Intervention

All therapeutic procedures were conducted between 1:00 PM and 5:00 PM, following a 6-hour fasting period. Under intravenous sedation, a gastroscope (EPX-2500, Fujifilm, Japan) was used for the initial evaluation of the esophagus and stomach. For the BTX-A injection (Letybo, Hugel, Korea; 100 IU in 10 ml saline), submucosal injections were performed using a 22-gauge, 1.9 mm thick, 180 cm long needle (Injectra, Medi-Globe, Germany).

We divided the individuals into four groups: Group 1, IGIB began 6 cm from the pyloric antrum at the 3, 6, 9, and 12 o'clock positions on the mucosa of the gastric body and fundus. Submucosal bullae were formed after each injection. This was repeated three times, moving 3 cm upwards towards the cardia each time. Each point received 2 ml (20 units) of Letybo, with a total of 20 injection points and an overall dose of 400 IU; Group 2, patients received the same procedure as Group 1, but with 15 injection points and a total dose of 300 IU of Letybo; Group 3, injections were administered closer to the pyloric antrum, starting 3 cm from it, at the 3, 6, 9, and 12 o'clock positions on the mucosa of the antrum and gastric body. The process was repeated three times, moving 3 cm upwards with a total of 20 injections and 400 IU of Letybo; Group 4, followed the same procedure as Group 3 but received 15 injections and a total dose of 300 IU of Letybo.

All patients were monitored for 1 hour after the endoscopic procedure. After the procedure, patients were prescribed a 1200 kcal liquid diet consisting of 15% protein, 35% lipid and 50% carbohydrate. This diet was personalized to individual preferences, sourced from available market options, ensuring variety while maintaining nutritional balance. All patients in this study received dietary evaluation and guidance from a dietitian both before and after the intervention. They were required to follow the liquid diet and maintain a stable lifestyle or body weight loss, and changes to concomitant medications were permitted, but only if maintained the instructions consistently for at least three months from baseline. Patients visited the clinic every 4 weeks for evaluations. Key measurements included body weight, BMI and body composition, which were taken at baseline, 1month and 3 months after the treatment. All measurements were conducted by an observer who was blinded to the specific treatment each patient received, which helps reduce bias.

### Statistical analysis

Descriptive statistics for the variables in the study are presented as mean and standard deviation, while categorical data were presented as counts. Baseline data for each variable were analyzed with one-way ANOVA. A mixed model was used

to analyze the effects of site, dose, and time for the variables. A p-value ≤0.05 was considered significant. Statistical analysis was performed with the program SPSS software (SPSS, Version 26, IBM, Chicago, Illinois)

## Results

A total of 103 patients met the inclusion criteria and were included in the study, with a mean age of 41.64 years (standard deviation = 8.93). Of these, 88 (85.44%) were female. The four groups represent different combinations of site (gastric body/fundus vs. gastric body/antrum) and dosage (400 IU and 300 IU). Measurements of patients' weight, BMI, and body fat were recorded at baseline, and followed up at 1 month and 3months post IGIB (Table 1). The obesity-related measurements change over time, and in this study, the trend showed a decrease over time (Fig 1). At baseline, there were no significant differences in bodyweight, BMI and body fat among groups (p = 0.326).

Given the repeated measurements and the within-patient correlations, a mixed-effects model was utilized for the analysis. To evaluate the effects of obesity-related measurements between different injection time x site, significant differences were found in bodyweight, BMI, and body fat across the various injection locations as adjusted for age, sex, and time (Table 2). Moreover, when accounting for the interactions between various injection sites and doses, the results revealed a statistically significant difference in bodyweight between different doses (p = 0.024). (Table 3) There were no significant effects observed for BMI; however, the sit-by-dose interactions were significant in body fat (p = 0.041). (Table 3). Three patients in Group 1, two in Group 2, and one in Group 4 experienced gastric discomfort during their stay in the recovery room. Four of them recovered with the use of proton pump inhibitors, while others improved spontaneously after a period of rest.

## Discussion

The primary outcomes of this study measured were the body weights, and the secondary outcomes were BMI and bioelectrical impedance analysis. The measurements were conveniently and easily taken by a research assistant at three

**Table 1. Characteristics of each individual group with initial weight, weight loss, body mass index, and body fat at baseline and follow-up periods.**

|  | Group 1 |  | Group 2 |  | Group 3 |  | Group 4 |  | p-value |
|---|---|---|---|---|---|---|---|---|---|
| **Age** | **42.45** | **± 9.73** | **41.65** | **± 9.04** | **45.60** | **± 4.67** | **39.16** | **± 7.93** | **0.441** |
| Gender |  |  |  |  |  |  |  |  | 0.236 |
| Female | 27 | (81.82%) | 42 | (91.30%) | 3 | (60.00%) | 16 | (84.21%) |  |
| Male | 6 | (18.18%) | 4 | (8.70%) | 2 | (40.00%) | 3 | (15.79%) |  |
| Weight |  |  |  |  |  |  |  |  |  |
| Baseline | 82.64 | ± 15.57 | 77.96 | ± 13.83 | 84.20 | ± 15.32 | 76.58 | ± 12.14 | 0.326 |
| 1M | 79.06 | ± 15.21 | 75.80 | ± 13.59 | 81.20 | ± 15.02 | 74.42 | ± 11.67 | 0.540 |
| 3M | 76.06 | ± 14.90 | 73.17 | ± 13.61 | 77.60 | ± 14.12 | 72.68 | ± 11.58 | 0.706 |
| BMI |  |  |  |  |  |  |  |  |  |
| Baseline | 30.76 | ± 4.64 | 29.43 | ± 4.17 | 29.20 | ± 4.02 | 28.84 | ± 3.00 | 0.364 |
| 1M | 29.36 | ± 4.53 | 28.59 | ± 4.06 | 28.00 | ± 4.42 | 28.00 | ± 2.87 | 0.652 |
| 3M | 28.27 | ± 4.36 | 27.57 | ± 4.28 | 26.60 | ± 4.22 | 27.95 | ± 3.95 | 0.808 |
| Body Fat |  |  |  |  |  |  |  |  |  |
| Baseline | 40.58 | ± 5.62 | 40.28 | ± 5.39 | 35.20 | ± 10.64 | 38.63 | ± 3.88 | 0.160 |
| 1M | 39.15 | ± 5.39 | 39.30 | ± 5.52 | 34.40 | ± 10.64 | 37.89 | ± 4.11 | 0.253 |
| 3M | 37.03 | ± 5.25 | 37.67 | ± 5.58 | 32.40 | ± 9.81 | 36.84 | ± 4.55 | 0.254 |

Continuous variables were presented as mean ± standard deviation (SD); categorical variables presented as n (%). Group 1: gastric fundus and body (BTX-A 400iu); Group 2: gastric fundus and body (BTX-A 300iu); Group 3: gastric antrum and body (BTX-A 400u); Group 4: gastric antrum and body (BTX-A 300iu).

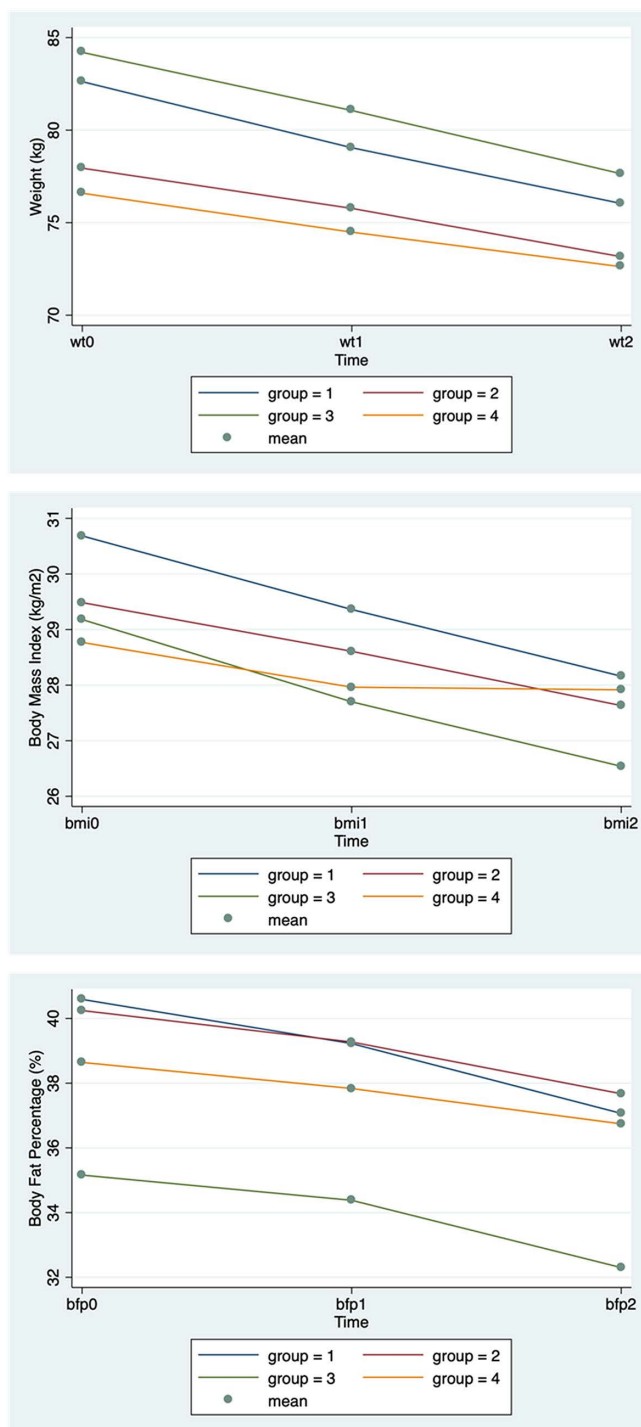

**Fig 1. The obesity-related measurements change over time.**

**Table 2. Type III tests of fixed effects for the mixed-effect model of time and site.**

| Outcome | Source | Numerator df | Denominator df | F | p-value |
|---|---|---|---|---|---|
| Weight | Intercept | 1 | 300.709 | 660.269 | <0.001 |
| | Time | 2 | 203.583 | 3.612 | .029 |
| | Site | 1 | 300.651 | 5.185 | .023 |
| | Time * Site | 2 | 203.583 | .043 | .958 |
| | Age | 1 | 300.625 | 5.406 | .021 |
| | Sex | 1 | 300.625 | 164.881 | <0.001 |
| | | | | | |
| BMI | Intercept | 1 | 300.363 | 737.426 | <0.001 |
| | Time | 2 | 196.524 | 3.667 | .027 |
| | Site | 1 | 300.236 | 5.026 | .026 |
| | Time * Site | 2 | 196.524 | .274 | .761 |
| | Age | 1 | 300.178 | .152 | .697 |
| | Sex | 1 | 300.178 | 62.468 | <0.001 |
| | | | | | |
| Body Fat | Intercept | 1 | 300.976 | 758.480 | <0.001 |
| | Time | 2 | 200.054 | 4.861 | .009 |
| | Site | 1 | 300.972 | 3.924 | .049 |
| | Time * Site | 2 | 200.054 | .193 | .825 |
| | Age | 1 | 300.969 | 2.404 | .122 |
| | Sex | 1 | 300.969 | 86.721 | <0.001 |
| | | | | | |

different time intervals from baseline to four- and 12-weeks post-procedure at the secondary health care clinic. We observed a statistically significant decrease in body weight, BMI, and body fat for time x site interactions: the significant reductions were observed in fundus and body injections across times. However, the site-by-dose interactions were not significant for any of the outcomes across the groups, with the exception of body fat. Furthermore, when considering the combined effects of different injection sites and doses, the analysis revealed a significant difference in body weight reduction between the varying dose levels.

The inconsistent results from prior studies highlight the need to examine key factors in the research, such as the dosage of BTX-A, the number and placement of injections…, etc. [6,13] Previous studies on BTX-A have shown a wide range of dosages, from 100 IU to 500 IU. [13] Interestingly, even at the highest dose, there was no observed impact on reducing body weight. The most effective dose for IGIB remains a point of contention. The effectiveness is likely influenced by the dose of BTX-A, the number and sites of injections administered. [8,13,15,16] Research has explored weight loss before and after BTX-A injection compared to a placebo group. Findings suggest that administering injections over a larger area, including the fundus or body rather than just the antrum, is linked to weight loss. [7,13,17] The aforementioned approach may explain in part to the significant results in our mixed-effects model of time x site analysis, which evaluate the effects of obesity-related measurements between different injection sites (fundus and body vs. antrum and body). Though injections into the gastric body were rarely part of earlier randomized controlled studies, [7] this area plays a role in temporarily storing food and partially digesting it chemically and mechanically. Gastric body injections may thus interfere with peristalsis and gastric juice secretion, potentially delaying gastric emptying time and reducing appetite. [18–21] Moreover, injection to fundus and body seems superior to injection into antrum and body in our results. [6]

**Table 3. Type III tests of fixed effects for the mixed-effect mode of time, site and dose.**

| Outcome | Source | Numerator df | Denominator df | F | p-value |
|---|---|---|---|---|---|
| Weight | Intercept | 1 | 296.757 | 4886.699 | <0.001 |
| | SITE | 1 | 296.757 | .022 | .881 |
| | Dose | 1 | 296.757 | 5.117 | .024 |
| | time | 2 | 200.069 | 2.010 | .137 |
| | SITE * Dose | 1 | 296.757 | .406 | .524 |
| | SITE * time | 2 | 200.069 | .003 | .997 |
| | Dose * time | 2 | 200.069 | .085 | .918 |
| | SITE * Dose * time | 2 | 200.069 | .010 | .990 |
| BMI | Intercept | 1 | 296.578 | 7463.170 | <0.001 |
| | SITE | 1 | 296.578 | 1.849 | .175 |
| | Dose | 1 | 296.578 | .210 | .647 |
| | time | 2 | 194.965 | 2.888 | .058 |
| | SITE * Dose | 1 | 296.578 | .917 | .339 |
| | SITE * time | 2 | 194.965 | .038 | .963 |
| | Dose * time | 2 | 194.965 | .255 | .775 |
| | SITE * Dose * time | 2 | 194.965 | .089 | .915 |
| Body Fat | Intercept | 1 | 296.998 | 7180.779 | <0.001 |
| | SITE | 1 | 296.998 | 12.368 | .001 |
| | Dose | 1 | 296.998 | 5.011 | .026 |
| | time | 2 | 198.116 | 3.156 | .045 |
| | SITE * Dose | 1 | 296.998 | 4.198 | .041 |
| | SITE * time | 2 | 198.116 | .066 | .936 |
| | Dose * time | 2 | 198.116 | .109 | .897 |
| | SITE * Dose * time | 2 | 198.116 | .006 | .994 |

Additionally, performing more than ten injections showed a stronger association with weight reduction in previous studies. [7] Our results demonstrated a significant weight loss in 400 IU (Group A and C, 20 injections) than in 300 IU (Group B and D, 15 injections). The physiological mechanism behind this effect remains unclear, contrasting with earlier studies that found higher doses of BTX-A were not linked to weight loss. Presumably, our improved efficacy might appear to result from increased intramuscular diffusion of the toxin. [6,8] Furthermore, combined fundus and body or antrum and body injections need more numbers of injections and dosage, [6] which have extended effects in the gastric function.

In addition to the pharmacological effects of BTX-A, dietary factors play a critical role in determining weight-loss outcomes. The calorie-restricted diet employed in this study likely exerted a synergistic effect with BTX-A administration. Specifically, dietary modification not only reduces overall caloric intake but may also attenuate interindividual variability in the therapeutic response to the toxin. Previous evidence indicates that the integration of intragastric interventions with structured nutritional guidance enhances weight reduction and promotes favorable metabolic adaptations [6,8,15,22]. Sustaining weight loss, however, requires a long-term commitment to lifestyle modification. Although BTX-A injections may facilitate an initial period of weight reduction, their effects are generally time-limited, and patients who revert to pre-intervention dietary behaviors remain vulnerable to weight regain. While the follow-up period in the present study was limited to three months, the findings underscore the importance of continued dietary counseling, regular physical activity, and behavioral support to preserve treatment benefits. The integration of these strategies is essential for transitioning from short-term weight reduction to durable weight maintenance and broader health benefits, including sustained metabolic improvements and a reduced risk of obesity-related complications.

This study is subject to several limitations. Firstly, patient enrollment was conducted consecutively, and allocation to groups was not randomized, introducing a potential for selection bias and an unequal distribution of participants across groups. Secondly, the dosage of BTX-A varied among studies, ranging from 100 to 500 IU; [13] however, in this analysis, only 300 IU and 400 IU were included for comparison. This decision was informed by some of the prior randomized studies demonstrating that doses equal to or under 200 IU did not significantly reduce body weights. [14] Thirdly, the study's short follow-up period may limit the ability to assess the long-term effectiveness of IGIB and reduce the generalizability of our findings to other contexts. Fourthly, the study did not include a placebo or standard-of-care control group. This limitation makes it difficult to distinguish the specific pharmacological effects of BTX-A from those of the structured 1200 kcal liquid diet and repeated dietary counseling, both of which can independently promote weight loss.

## Conclusion

The findings of this study provide preliminary evidence supporting the potential efficacy of endoscopic intragastric botulinum toxin A (BTX-A) injections in the management of obesity. Key factors contributing to successful outcomes include the use of an optimal BTX-A dose (400 IU compared to 300 IU) and the administration of multiple injections (20 versus 15) across a broad gastric area, in conjunction with a structured dietary protocol. This method demonstrated a favorable safety profile with minimal adverse effects, suggesting it may represent a viable treatment option for patients who are not suitable candidates for surgical intervention. Nevertheless, the absence of a placebo or standard-of-care control group limits the ability to isolate the pharmacological effects of BTX-A from concurrent dietary interventions. Future randomized, placebo-controlled trials are warranted to confirm and validate these observed effect sizes.

## Supporting information

**S1 Data. Raw data from all patients presented in the present study.**
(XLSX)

## Author contributions

**Conceptualization:** Cheng-Hung Chiu.

**Formal analysis:** I-Wen Penn, Cheng-Di Chiu.

**Methodology:** Tien-Yow Chuang, I-Wen Penn.

**Project administration:** Tien-Yow Chuang, I-Wen Penn.

**Writing – original draft:** Cheng-Di Chiu.

**Writing – review & editing:** Tien-Yow Chuang.

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
