## [Decision Letter · Decision Letter 0]

10 Apr 2025

Dear Dr. Chiu,

Thank you for submitting your manuscript to PLOS ONE. After careful consideration, we feel that it has merit but does not fully meet PLOS ONE’s publication criteria as it currently stands. Therefore, we invite you to submit a revised version of the manuscript that addresses the points raised during the review process.

We look forward to receiving your revised manuscript.

Kind regards,

Roland Eghoghosoa Akhigbe

Academic Editor

PLOS ONE

Journal Requirements:

3. We note that there is identifying data in the Supporting Information file “Supporting information.xlsx”. Due to the inclusion of these potentially identifying data, we have removed this file from your file inventory. Prior to sharing human research participant data, authors should consult with an ethics committee to ensure data are shared in accordance with participant consent and all applicable local laws.

-Location data

Reviewers' comments:

Reviewer's Responses to Questions

**Comments to the Author**

1. Is the manuscript technically sound, and do the data support the conclusions?

Reviewer #1: Yes

Reviewer #2: Yes

2. Has the statistical analysis been performed appropriately and rigorously?

Reviewer #1: Yes

Reviewer #2: I Don't Know

3. Have the authors made all data underlying the findings in their manuscript fully available?

Reviewer #1: Yes

Reviewer #2: Yes

4. Is the manuscript presented in an intelligible fashion and written in standard English?

Reviewer #1: Yes

Reviewer #2: Yes

Reviewer #1: Chuang etal conduct a retrospective study to investigate the efficacy of two different doses, multi-gastric

sites botulinum toxin A injections for weight control in patients with obesity. I listed my concerns as follows

Major Concerns

1. Diagnosis of the pricipants. In the manuscript, authors stated that pariticpants with BMI at least 25kg/M2 otherwise with complications were allowed to be enrolled. But please make it clear, whether these participants could be diagnosed with obesity or overweight or not.

2. Study design. Please clarify whether these pariticpants had maintained stable life style or weight (change less than 5%) for at least three months or even longer at baseline or not. Also, please also clarify whether the change of concomitant medicine or lifestyle change is allowed throughout the study.

3. Safety data was missed. Authors just simply described the efficacy data while didn't touch any adverse event like GI side effect which I believe is very important to evalute the treatment regimen together with efficacy data.

Minor

Overspeculation.

In the conclusion part, the authors claimed that this is a strong evidence, which is obviously inappropriate since it is not a RCT. Please correct.

Reviewer #2: This interesting study evaluated the outcome of intragastric BTX-A injections in 4 groups of patients receiving endoscopic injections at varying doses (300 IU or 400 IU) and sites (fundus and body or antrum and body). The results showed significant reductions in body weight, BMI, and body fat with gastric fundus and body injections. The authors concluded that endoscopic intragastric BTX-A injections effectively reduce body weight, BMI, and body fat, particularly with injections in the fundus and body regions.

The study was well conducted, and the article is well constructed. I have a few comments for the authors:

-It is mentioned that patients were required to follow a 1200 kcal liquid diet for 12 weeks, which is quite a long time. Did patients have any difficulties with the liquid diet? Did all patients keep their diet? Was there a recommended diet for patients after 12 weeks? Did you follow up with patients after changing the liquid diet? A liquid diet alone can result in significant weight loss and changing the diet after 12 weeks may lead to weight gain. A longer follow-up of patients after switching to a regular diet would be better.

-Were there any short-term or long-term complications after the procedure?

**Do you want your identity to be public for this peer review?** For information about this choice, including consent withdrawal, please see our Privacy Policy

Reviewer #1: No

Reviewer #2: No

---

## [Author Response · Author response to Decision Letter 1]

26 Apr 2025

Academic Editor

Professor Roland Eghoghosoa Akhigbe

Plos One Editorial Office

Re: “Evaluating the Efficacy of Intragastric Botulinum Toxin A Injections with Two Different Quantities and Sites for Obesity “ PONE-D-24-57405

”

Dear Professor Akhigbe

We resubmitting our manuscript entitled ”Evaluating the Efficacy of Intragastric Botulinum Toxin A Injections with Two Different Quantities and Sites for Obesity”

”

The manuscript has been revised in response to the reviewers’ comments and we hope that it is now suitable for publication.

The major changes we have made are the following:

The article has been rewritten with the aid of a native English speaker. We believe that this removal of unintentional ambiguities of wording has answered many of the reviewers’ questions.

We have rewritten the method & materials, results and conclusion sections according to your comments.

We have put on the accompanying pages our responses to each of your requests, as well as our responses to the specific comments of each reviewer.

Sincerely,

Cheng-Di Chiu, MD, PhD Department of Neurosurgery, China Medical University & Hospital, Taichung, Taiwan

Email: cdchiu4046@gmail.com

Here are our responses (by item) to the comments of reviewers. We sub-itemized the questions in order to answer them more specifically.

Please include your full ethics statement in the ‘Methods’ section of your

manuscript file. In your statement, please include the full name of the IRB or

ethics committee who approved or waived your study, as well as whether or

not you obtained informed written or verbal consent. If consent was waived for

your study, please include this information in your statement as well.

Ans: We followed your suggestion and re-wrote this part in Material and Methods section (Lines 97-101).

Ans: Thank you for your notice. We have carefully reviewed the Supporting Information file and have removed all personally identifiable information, including any potentially identifying columns. All data have been anonymized in accordance with participant consent and ethical guidelines. The revised and fully anonymized dataset has been re-uploaded.

Reviewer 1

In the manuscript, authors stated that participants with BMI at least 25kg/M2 otherwise with complications were allowed to be enrolled. But please make it clear, whether these participants could be diagnosed with obesity or overweight or not.

Ans: Agreed. We re-wrote this part in Lines 105-108 (highlighted) to make it more clear.

Study design. Please clarify whether these participants had maintained

stable life style or weight (change less than 5%) for at least three months or

even longer at baseline or not. Also, please also clarify whether the change of

concomitant medicine or lifestyle change is allowed throughout the study.

Ans: Yes, we follow your suggestions and re-edit this paragraph in Lines 134-139 (highlighted).

Safety data was missed. Authors just simply described the efficacy data

while didn’t touch any adverse event like GI side effect which I believe is very important to evalute the treatment regimen together with efficacy data.

Ans: We agree with your opinion and add some descriptions on the adverse events. However, six patients sustained mild immediate post-intervention GI upset without any long-term side effects. Lines 166-168 (highlighted).

In the conclusion part, the authors claimed that this is a strong evidence,

which is obviously inappropriate since it is not a RCT.

Ans: Agreed. We have deleted the word “ strong” in the original sentence (Line 234).

Reviewer 2

It is mentioned that patients were required to follow a 1200 kcal liquid diet for

12 weeks, which is quite a long time. Did patients have any difficulties with the

liquid diet? Did all patients keep their diet.

Ans: This diet was personalized to individual preferences, sourced from available market options, ensuring variety while maintaining nutritional balance. Therefore, all participants can keep their diet. We have re-edit this part in Lines 134-139.

Was there a recommended diet for patients after 12 weeks? Did you follow up with patients after changing the liquid diet? A liquid diet alone can result in significant weight loss and changing the diet after 12 weeks may lead to weight gain. A longer follow-up of patients after switching to a regular diet would be better.

Ans: It is very probable that in our participants a combined effect of reductive diet and toxin was observed. We agree with you. Although, all patients in this study received dietary evaluation and guidance from a dietitian both before and after the intervention, changing the diet after 12 weeks may lead to weight gain. Obesity is often closely linked to lifestyle choices. A sedentary lifestyle—characterized by minimal physical activity and prolonged sitting—combined with unhealthy eating habits, such as frequent consumption of high-calorie, processed foods, contributes significantly to weight gain and obesity.

However, our study focused on the effects of different doses and sites of intragastric BTX-A injections at this stage. We are happy to follow and help the overweight and obese patients in the long run to switch to a regular diet in our future study.

Were there any short-term or long-term complications after the procedure?

Ans: Six patients sustained mild immediate post-intervention GI discomforts without any long-term side effects. Lines 166-168 (highlighted).

---

## [Decision Letter · Decision Letter 1]

25 Aug 2025

Dear Dr. Chiu,

Thank you for submitting your manuscript to PLOS ONE. After careful consideration, we feel that it has merit but does not fully meet PLOS ONE’s publication criteria as it currently stands. Therefore, we invite you to submit a revised version of the manuscript that addresses the points raised during the review process.

We look forward to receiving your revised manuscript.

Kind regards,

Eshetie Melese Birru, PhD

Academic Editor

PLOS ONE

Journal Requirements:

Reviewers' comments:

Reviewer's Responses to Questions

**Comments to the Author**

Reviewer #2: All comments have been addressed

2. Is the manuscript technically sound, and do the data support the conclusions?

Reviewer #2: Yes

3. Has the statistical analysis been performed appropriately and rigorously?

Reviewer #2: I Don't Know

4. Have the authors made all data underlying the findings in their manuscript fully available?

Reviewer #2: Yes

5. Is the manuscript presented in an intelligible fashion and written in standard English?

Reviewer #2: Yes

Reviewer #2: Dear authors,

Thank you for your response and for addressing the comments. I believe it would be beneficial to discuss the combined effect of a reductive diet and toxins in the discussion section, as well as the importance of long-term lifestyle choices in maintaining weight.

Additionally, if you have any considerations about the diet after three months, I would suggest including them in the manuscript.

**Do you want your identity to be public for this peer review?** For information about this choice, including consent withdrawal, please see our Privacy Policy

Reviewer #2: No

---

## [Author Response · Author response to Decision Letter 2]

4 Sep 2025

Here are our responses (by item) to the comments of reviewer 2

Thank you for your response and for addressing the comments. I believe it would be beneficial to discuss the combined effect of a reductive diet and toxins in the discussion section, as well as the importance of long-term lifestyle choices in maintaining weight.

Ans: We sincerely appreciate your insightful comments and constructive suggestions. Following your recommendation, we have expanded the Discussion section (Line 217-230) to address the combined effect of a calorie-restrictive diet and botulinum toxin A, as well as the importance of long-term lifestyle modifications in sustaining weight loss. Specifically, we emphasized that while intragastric botulinum toxin A injections may facilitate initial weight reduction, durable outcomes are contingent upon continued adherence to healthy dietary patterns and behavioral strategies over time. We also underscored that the supportive role of a calorie-restrictive, nutritionally balanced diet is likely synergistic with the pharmacological intervention, as reported in recent studies.

Additionally, if you have any considerations about the diet after three months, I would suggest including them in the manuscript.

Ans: Thank you for your suggestions. We incorporated remarks regarding dietary considerations beyond the three-month follow-up. Although our study period was limited to 12 weeks, we acknowledged the potential for weight regain if patients revert to pre-intervention eating habits. Accordingly, we highlighted the necessity of structured dietary counseling and lifestyle interventions to achieve and maintain long-term success.

---

## [Decision Letter · Decision Letter 2]

21 Oct 2025

Dear Dr. Chiu,

publication criteria  and not, for example, on novelty or perceived impact.

We look forward to receiving your revised manuscript.

Kind regards,

Eshetie Melese Birru, PhD

Academic Editor

PLOS ONE

Journal Requirements:

Additional Editor Comments (if provided):

Although the study explores different dose–site combinations of intragastric botulinum toxin A (BTX-A), it lacks a placebo or standard-of-care control group.

This omission makes it impossible to isolate the pharmacological effect of BTX-A from that of the structured 1200 kcal liquid diet and repeated dietary counselling—all of which independently induce weight loss.

Without a control arm:

The reported reductions in body weight, BMI, and body fat could reflect dietary restriction rather than toxin efficacy.

The mixed-model analysis comparing “site” and “dose” groups assumes BTX-A is effective, yet there is no baseline-to-placebo benchmark to confirm this.

Prior RCTs have yielded heterogeneous results, emphasizing the need for clear differentiation between toxin and behavioral effects.

Required correction/comment:

The authors must explicitly acknowledge this as a major limitation and temper the conclusion that “endoscopic intragastric BTX-A injections effectively reduce body weight, BMI, and body fat.”

Alternatively, if historical or concurrent controls were available, these should be clearly described, with corresponding data or justification for comparability.

Future trials should include a randomized placebo-controlled design to validate the observed effect sizes.

Reviewers' comments:

Reviewer's Responses to Questions

**Comments to the Author**

Reviewer #2: All comments have been addressed

2. Is the manuscript technically sound, and do the data support the conclusions?

Reviewer #2: Yes

3. Has the statistical analysis been performed appropriately and rigorously?

Reviewer #2: I Don't Know

4. Have the authors made all data underlying the findings in their manuscript fully available?

Reviewer #2: Yes

5. Is the manuscript presented in an intelligible fashion and written in standard English?

Reviewer #2: Yes

Reviewer #2: (No Response)

**Do you want your identity to be public for this peer review?** For information about this choice, including consent withdrawal, please see our Privacy Policy

Reviewer #2: No

---

## [Author Response · Author response to Decision Letter 3]

3 Nov 2025

Although the study explores different dose–site combinations of intragastric botulinum toxin A (BTX-A), it lacks a placebo or standard-of-care control group. This omission makes it impossible to isolate the pharmacological effect of BTX-A from that of the structured 1200 kcal liquid diet and repeated dietary counselling—all of which independently induce weight loss. Without a control arm: The reported reductions in body weight, BMI, and body fat could reflect dietary restriction rather than toxin efficacy. The mixed-model analysis comparing “site” and “dose” groups assumes BTX-A is effective, yet there is no baseline-to-placebo benchmark to confirm this. Prior RCTs have yielded heterogeneous results, emphasizing the need for clear differentiation between toxin and behavioral effects.

Required correction/comment: The authors must explicitly acknowledge this as a major limitation and temper the conclusion that “endoscopic intragastric BTX-A injections effectively reduce body weight, BMI, and body fat.” Alternatively, if historical or concurrent controls were available, these should be clearly described, with corresponding data or justification for comparability. Future trials should include a randomized placebo-controlled design to validate the observed effect sizes.

Response: In accordance with this recommendation, we have expanded the Discussion section (Lines 239–242) to explicitly acknowledge this limitation, emphasizing that the current study design does not permit a clear distinction between the pharmacological effects of endoscopic intragastric BTX-A injections and the concurrent influence of caloric restriction and dietary counseling.

Furthermore, we have revised the Conclusion section (Lines 251–254) to indicate that future investigations will adopt a randomized, placebo-controlled design to confirm and validate the observed effect sizes.

---

## [Editor Report · Decision Letter 3]

4 Dec 2025

Evaluating the Efficacy of Intragastric Botulinum Toxin A Injections with Two Different Quantities and Sites for Obesity

PONE-D-24-57405R3

Dear Dr. Cheng-Di,

We’re pleased to inform you that your manuscript has been judged scientifically suitable for publication and will be formally accepted for publication once it meets all outstanding technical requirements.

Kind regards,

Eshetie Melese Birru, PhD

Academic Editor

PLOS ONE
---

## [Editor Report · Acceptance letter]

PONE-D-24-57405R3

PLOS One

Dear Dr. Chiu,

I'm pleased to inform you that your manuscript has been deemed suitable for publication in PLOS One. Congratulations! Your manuscript is now being handed over to our production team.

Kind regards,

on behalf of

Dr. Eshetie Melese Birru

Academic Editor

PLOS One